# Understanding the Geographic Patterns of Closely-Related Species of *Paspalum* (Poaceae) Using Distribution Modelling and Seed Germination Traits

**DOI:** 10.3390/plants12061342

**Published:** 2023-03-16

**Authors:** Nicolás Glison, David Romero, Virginia Rosso, José Carlos Guerrero, Pablo Rafael Speranza

**Affiliations:** 1Departamento de Biología Vegetal, Facultad de Agronomía, Universidad de la República, Avenida Eugenio Garzón 780, Montevideo 12900, Uruguay; 2Laboratorio de Desarrollo Sustentable y Gestión Ambiental del Territorio, Instituto de Ecología y Ciencias Ambientales, Facultad de Ciencias, Universidad de la República, Iguá 4225, Montevideo 11400, Uruguay; 3Grupo Biogeografía, Diversidad & Conservación, Departamento Biología Animal, Facultad de Ciencias, Universidad de Málaga, Campus de Teatinos s/n, 29071 Málaga, Spain; 4Cátedra de Botánica Sistemática, Facultad de Agronomía, Universidad de Buenos Aires, Avenida San Martín 4453, Buenos Aires C1417DSE, Argentina

**Keywords:** favourability function, local adaptation, native grasses, regeneration traits, seed dormancy, specialist species, South America

## Abstract

The sexual species of the Dilatata complex (*Paspalum dasypleurum*, *P. flavescens*, *P. plurinerve*, *P. vacarianum,* and *P. urvillei*) are closely related phylogenetically and show allopatric distributions, except *P. urvillei*. These species show microhabitat similarities and differences in germination traits. We integrated species distribution models (SDMs) and seed germination assays to determine whether germination divergences explain their biogeographic pattern. We trained SDMs in South America using species’ presence–absence data and environmental variables. Additionally, populations sampled from highly favourable areas in the SDMs of these species were grown together, and their seeds germinated at different temperatures and dormancy-breaking conditions. Differences among species in seed dormancy and germination niche breadth were tested, and linear regressions between seed dormancy and climatic variables were explored. SDMs correctly classified both the observed presences and absences. Spatial factors and anthropogenic activities were the main factors explaining these distributions. Both SDMs and germination analyses confirmed that the niche of *P. urvillei* was broader than the other species which showed restricted distributions, narrower germination niches, and high correlations between seed dormancy and precipitation regimes. Both approaches provided evidence about the generalist-specialist status of each species. Divergences in seed dormancy between the specialist species could explain these allopatric distributions.

## 1. Introduction

The grasslands of midlatitude lowland regions of South America show an unusually high biodiversity. In particular, the grassland in the southeastern part of South America is one of the largest and most diverse in the world [1]. In these regions, recently diverged species, populations, and entities forming species complexes have been described in several taxa. This high intraspecific variability has been attributed to repeated past range fluctuations and fragmentation into multiple climatic refugia [2,3,4,5,6]. However, biodiversity in such regions may be threatened due to land-use changes, mainly due to the intensification of agriculture and forestry in the few last decades [1,7] and the absence of adequate conservation policies [8].

The genus *Paspalum* (Poaceae) is a highly diverse taxon in the South American grasslands [9]. Within *Paspalum*, the Dilatata species complex is an allopolyploid group of warm-season grasses composed of several apomictic taxa, such as the pentaploid *Paspalum dilatatum* Poir., which has been extensively studied as a forage crop [10,11]. This group also contains five sexual species: *P. dasypleurum* Kunze ex. Desv., *P. flavescens* (Roseng., Arrill. and Izag.) Speranza and G. H. Rua, *P. plurinerve* Quarin, Valls and V. C. Rosso, *P. vacarianum* Valls and V. C. Rosso, and *P. urvillei* Steud. [12]. The apomictic species of the Dilatata group encompass rather indefinite groupings of interspecific hybrids [13]. However, despite their close phylogenetic relationship, the five sexual species included in this group can be considered very well delimited and recently diverged independent evolutionary units [14,15]. These sexual species are highly autogamous [16] and, although hybrids can be obtained artificially, mostly no gene flow has been observed in nature among them [17]. 

The geographical location of the sexual Dilatata species has been recently reported, however, its distribution has not been biogeographically analysed. Four of the five species (*P. dasypleurum*, *P. flavescens*, *P. plurinerve*, and *P. vacarianum*) show a restricted and mutually exclusive geographical distribution, however, *P. urvillei* shows a wider continuous distribution co-occurring in its native range with the last three species mentioned above [12,18]. The reported distributions suggest that *P. urvillei* has a generalist habitat behaviour, which translates into its ability to occupy diverse environments and larger areas at the ecological scale. In contrast, the other four species are closely associated with more specific regions and are probably more specialist [19]. Despite this, all the sexual Dilatata species occupy similar microhabitats with high light and soil water availability and are frequently found on roadsides and disturbed ranges [12,16,20]. 

The distribution of a plant species is determined by historical and environmental factors and its interaction with other species. Species distribution models (SDMs) have been extensively used as an approach to understanding the potential distribution of a species and to test biogeographical, ecological, and evolutionary hypotheses [21,22,23,24,25,26]. SDMs help identify drivers that determine the favourable regions for a species and the habitat characteristics that define their ecological niche breadth [24,27,28]. A broadly used SDM method consists of correlative models that use logistic regressions to link species’ presence-absence data with environmental and geographic data. [29,30]. Among these SDM methods, the favourability function is suitable for SDM comparisons between species with different prevalences in the same study area through fuzzy logic tools [31,32,33,34].

Crucial ecological processes that determine geographic distribution at the scale of environments or habitats, such as progeny dispersion, physiological responses to environmental stresses and biotic interactions, are not easy to incorporate explicitly into the SDMs [35,36]. However, integrating relevant plant functional traits as a species performance currency could be informative for understanding the mechanistic basis of the adaptation associated with the described distributions [37,38]. Functional traits are measurable individual-level features that interact with environmental and ecological factors [39]. Ecological hypotheses have been tested mainly with functional vegetative traits, frequently overlooking regeneration traits [40]. In spite of this, vegetative and regeneration traits may show contrasting relationships with environmental factors [41,42].

The regeneration niche concept states that plant species occur in habitats where seed germination and seedling establishment are possible [43,44,45]. Seed germination under a range of conditions, mainly different temperatures, has been used to test the regeneration niche hypothesis [46,47,48,49,50]. Seed germination responses are controlled by seed dormancy, a quantitative and adaptable mechanism that inhibits germination until specific environmental requirements are met. It can be estimated by evaluating germination in various temperatures and dormancy-breaking factors [51,52]. The ability to germinate in a wider temperature range often reflects reduced dormancy and a broader germination niche, which may be associated with a generalist habitat behaviour [53,54,55]. Seed dormancy mechanisms and germination responses to temperature are often highly conserved phylogenetically [56,57,58]. However, divergence in these traits among similar species is usual even across fine-scale environmental variations, giving adaptive germination responses driven by both genetic selection and phenotypic plasticity influenced by maternal effects [59,60,61]. 

The sexual species of the Dilatata complex share several functional vegetative traits with the very widespread and better studied *P. dilatatum*. They are perennial warm-season C4 grasses with very high leaf frost tolerance [62,63,64,65] and minor differences in growth habits [12,66]. Except for *P. dasypleurum*, no major differences in development or vegetative persistence have been reported for these species in their reciprocal locations when transplanted. Despite this, divergences can be relevant among regeneration traits. *Paspalum flavescens* has been shown to exhibit a stronger seed dormancy than *P. plurinerve*, which occurs further north [67,68]. However, information on germination for the rest of the sexual Dilatata species is scarce [69]. In general, seed germination in the species of the Dilatata complex occurs in relatively high temperatures. Germination can be improved by seed dormancy-breaking factors such as cold stratification, nitrate addition and alternating temperatures [67,70,71]. Each of these germination responses can be linked to ecological and habitat preferences. For example, the germination proportion achieved after cold stratification reflects the extent of the cold requirement during winter to germinate in spring [72]. Also, the response to nitrate is a seed gap-detection mechanism and can be related to the preference for disturbed places [51,73].

In this work, we integrated species’ distribution modelling with the study of a regeneration trait to understand the geographical range occupied by the sexual Dilatata species, and to explain the current biogeographical pattern of this group. First, we applied favourability models based on presence-absence data and a wide range of predictive variables (spatial, topographic, climatic, land use, and anthropogenic activities) to determine the favourable area of each species and identify key drivers determining their distributions. To determine whether environmental favourability is related to germination traits, we sampled genotypes (inbred lines) of the five species from locations with a high likelihood of being located in the respective favourable areas. We grew them in a common garden experiment and produced seeds to study germination under a range of alternating temperatures, nitrate addition, and stratification treatments for two years. Seed dormancy and germination niche breadth were estimated based on the germination results. We evaluated the differences in seed dormancy among the species and performed regressions between germination and environmental variables for each population. The ecological and biogeographical insights brought by each approach were discussed.

## 2. Results

### 2.1. Explanatory Variables and Favourable Areas for Sexual Dilatata Species

The favourability results were congruent with the territories occupied by the observed presences for each species (Figure 1). The spatial variable (Ysp) was relevant across the models of the five species, and it was the most important predictor for the distribution of *P. flavescens* and *P. plurinerve* (Table 1). Meanwhile, the urban cover was the most relevant predictor for *P. dasypleurum* and *P. urvillei*, and the minimum temperatures in the coldest month (Bio6) for *P. vacarianum*. For all the species analysed, at least one relevant model variable was related to human activities (Table 1). All the explanatory variables of each model do not show multicollinearity (variance inflation factor < 2). All the sites where the genotypes used for the germination experiments were sampled appeared in hexagons classified as of maximum favourability (F ≥ 0.8), except for the northernmost point for *P. urvillei* (red triangles in Figure 1). The area under the receiving operating characteristic curve (AUC) was higher than 0.9 for all the models, which is interpreted as outstanding discrimination ability (see Table A2). Moreover, the values obtained for the classification parameters (sensitivity, specificity, correct classification rate and true skill statistic) were always greater than 0.9 in all models except for *P. urvillei* (Table A2). 

### 2.2. Germination Treatments

The harvested seeds were used to perform two experiments to analyse the germination behaviour of these species. Sufficient seeds were collected for all genotypes. Particularly, the number of seeds harvested for *P. urvillei* genotypes was two- to sixfold than that of the other species in all harvest times. Experiment one was set to assess germination after stratification treatments. Experiment two, on the other hand, was designed to assess germination in a range of alternating temperatures with and without the addition of nitrate to determine the thermal amplitude of germination in the presence or not of a dormancy-breaking substance. The final germination proportion (FGP) showed significant differences among treatment levels and species in both germination experiments (Table 2). However, the uncertainty of germination (UG) and germination timing variables (mean germination section for experiment one and mean germination time for experiment two) showed inconsistent differences among species through treatments or no differences at all (see Table A4). Cold stratification resulted in a higher FGP than nonstratified seeds (*p* < 0.05), except for *P. dasypleurum*, which reached germination proportions lower than 10% in all treatment levels in experiment one. Warm stratification showed lower FGP than cold stratification for *P. flavescens*, *P. plurinerve*, and *P. vacarianum,* though not for *P. urvillei* (Table 2a). For experiment two, germination without nitrate was almost nil in colder temperatures (10/20 °C) and very low at 15/25 °C, except for *P. urvillei* and *P. plurinerve*, which achieved 75% and 21%, respectively. The addition of nitrate allowed a higher FGP in almost all temperatures and species (Table 2b). Germination with nitrate was close to 100% at high temperatures (20/30 and 25/35 °C) for all the species (except *P. dasypleurum*) and at 15/25 °C for *P. urvillei* and *P. plurinerve* (Table 2b). Combining high temperatures with nitrate also retrieved higher germination synchronicity (lower UG, Table A4).

The ranking from higher to lower FGP in almost all treatment levels for both experiments was *P. urvillei*, *P. plurinerve*, *P. vacarianum*, *P. flavescens*, and *P. dasypleurum*. This ranking was observed for germination ability (GA) and germination evenness (GE), two indexes used to estimate both seed dormancy level and germination niche breadth (Figure 2). The variance of GA and GE was higher among species than among genotypes within species or among harvest times (Table A5). The distribution of the variability in the extent, timing, and synchronicity of germination among genotypes and harvest times is depicted in the results of the PCA (Figure A2a). The first principal component (PC1) explained 32.2% of germination variability, and the five species showed a clear gradient along it. PC1 was positively correlated with FGP for seeds with warm stratification and without stratification (Figure A2b), which are restricting conditions for germination in the studied species. Also, PC1 is negatively correlated with UG at 20/30 and 25/35 °C with nitrate (Figure A2b), conditions that tend to increase synchronicity (lower UG). Therefore, the PC1 summarized several germination attributes, and its score is inversely related to seed dormancy.

### 2.3. Relationships between Germination and Environmental Variables 

Despite minor intraspecific differences, all genotypes of *P. urvillei* at all harvest times attained higher FGP, lower UG, and lower germination timing than the other species under most conditions assayed, which is reflected in higher PC1 scores (Figure A2). These results suggest a low seed dormancy for all *P. urvillei* genotypes, meaning that germination in this species is less environmentally constrained regardless of the large distances and differences in climatic features among their occurrence sites (Figure 1). This idea was further supported when the PLSR including *P. urvillei* genotypes explained a low proportion of the variability of both response (<62%) and predictor (<82%) variables for all the germination variables (Table 3a). Also, linear regressions between germination and environmental variables (those which achieved a VIP > 1 in PLSR) were significant (*p* < 0.05), however, they showed low adjustment (*R^2^* < 0.5) when all species were considered (Table A6). 

On the other hand, PLSR without *P. urvillei* explained a higher proportion of the variability of response (>94%) and predictor (>87%) variables (Table 3b). The set of environmental variables with VIP > 1 was almost the same for the three germination variables, all of which were climatic variables. Precipitation in the driest or warmest periods (Bio14, Bio17, and Bio18, see Table A1) showed an orthogonal relationship with the mean temperature in the colder quarter (Bio11) and annual potential evapotranspiration (ETP-An) (Figure A3). The correlation between the values of Bio14 and Bio17 was high (*r* = 0.999), and both were correlated with Bio18 (*r* = 0.90).

Significant linear regressions were obtained for all germination variables using the climatic variables with VIP > 1 after PLSR without *P. urvillei* genotypes (Table A6). All regressions achieved the highest adjustments when precipitations in the drier quarter (Bio17) were used as the predictor variable (Figure 3), and similar results were obtained using precipitation in the drier month (Bio14) or the warmest quarter (Bio18) (Table A6). Regressions using temperature-based variables, such as Bio11 or ETP-An, yielded significant, but lower, adjustments (*R^2^* < 0.7) (Table A6). Despite the high adjustment in regressions achieved by Bio17, this variable was not different between the locations where samples of *P. vacarianum* and *P. plurinerve* were collected, however, differences in the temperature in the coldest quarter (Bio11) were found (Figure 3). These two species did not show significant differences in GA or GE (Figure 2), though they showed consistent differences in germination at 15/25 °C even with nitrate, where *P. vacarianum* showed lower FGP than *P. plurinerve* (Table 2b).

## 3. Discussion

### 3.1. Explanatory Variables of Distribution Models and Congruence with Germination Responses

The distribution models of each species yielded a high favourability area which is highly coincident with the observed distribution range of each species. Some species had topographic or climatic variables as significant predictors for their models (slope for *P. dasypleurum*, mean temperature of colder months (Bio6) for *P. vacarianum*), which are representative factors of the environment in their respective sites. However, spatial and human activity variables were the most relevant distribution model predictors for all of the species. The relevance of spatial variables in the distribution of plant species is not related to germination traits, though it is often associated with low seed dispersal, which is another important regeneration trait [21,74]. The studied species exhibit autochorus dispersal mechanisms, similar to other grasses such as *Panicum maximum* L. [75]. Seeds fall near the mother plants by gravity, which does not enhance dispersal. Species with lower dispersal tend to show seed dormancy, which can be seen as a tradeoff between spatial and temporal dispersion [76].

On the other hand, the importance of human activities on distribution suggests that ruderal environments may be particularly suitable for these species. The preference of these species for disturbed environments, including those caused by anthropogenic factors, has been previously noted [12,16,77]. Soil disturbance may improve the germination of some species by increasing the levels of dormancy-breaking factors such as light, alternating temperatures or nitrate [51]. It has been shown that disturbances such as flooding and grazing lead to increased seedling emergence of *P. dilatatum* [20]. Coincidently, in the germination experiments, we obtained a positive response to nitrate addition for all the Dilatata species evaluated, and almost 100% of the seeds germinated with high synchronicity (except for *P. dasypleurum*) when nitrate addition and high alternating temperatures were combined. Nitrate in the soil is considered an environmental cue for the absence of vegetation or soil disturbance. The nitrate levels in the soil are greater in vegetation gaps than under undisturbed vegetation due to the nitrate uptake of established plants [73]. Also, soil disturbances induce an increase in nitrification [78]. The increase in seed germination with nitrate may be related to a preference for disturbed soils [51,79]. Thus, this opportunistic germination response to nitrate is congruent with the importance of human activities highlighted as one of the main explanatory predictors of the distribution of these species. 

### 3.2. Habitat Generalist and Specialist Behaviours and Germination Niche Breadth

Collection data and favourability models clearly differentiate between the broader geographical area covered by *P. urvillei* and the rather restricted and allopatric areas shown by the other four sexual Dilatata species. The large geographical area with high favourability modelled for *P. urvillei* encompasses diverse environments, suggesting a habitat generalist behaviour [19]. Also, SDM evaluation parameters were lower for *P. urvillei* than for the other species (see Table A2), which is characteristic of generalist behaviour [27,28]. For such species, some functional traits are expected to show a broader niche breadth to occupy several sites along an environmental gradient [80]. The wider germination niche showed by *P. urvillei* (i.e.: high germination proportion in almost all conditions assayed and higher PC1 scores, GA and GE) may indeed be the functional trait that explains its generalist behaviour [54]. In addition, the seed production of *P. urvillei* was higher than that of the other sexual Dilatata species. Its large seed production, combined with a broader germination niche, may explain why *P. urvillei* has been widely reported outside its native range as a weed for summer crops and roadsides in subtropical regions [81,82,83].

The favourability values of the other sexual Dilatata species (*P. dasypleurum*, *P. flavescens*, *P. plurinerve*, and *P. vacarianum*) were high in more restricted areas associated with a given environment, which suggests a more habitat specialist-like behaviour. In such cases, functional traits often show a smaller niche breadth and more prominent local adaptation features [84]. The four specialist species showed a narrower germination niche than *P. urvillei*. Moreover, although the intraspecific variability should be further regarded, a cline in germination responses was apparent through these species. The southernmost species (*P. dasypleurum* and *P. flavescens*) showed a higher seed dormancy than the northernmost species (*P. plurinerve* and *P. vacarianum*), which means higher cold stratification and nitrate requirements to release seed dormancy, and more restrictions to germinate in lower temperatures. A positive association between seed dormancy and latitude is often reported for transitional regions such as wet subtropical and temperate climates [85]. However, this relationship cannot be generalized [86]. In higher latitudes, the odds of adverse events such as frost or soil–water deficit are higher, even in spring, which is a good moment for warm-season grasses to germinate and establish. For these species, a higher seed dormancy allows a longer spreading in the germination time of the seed population, distributing the seedling death risk in a highly fluctuating environment [87].

### 3.3. Correlation between Germination Traits and Environmental Gradient

Precipitation during the driest or warmest part of the year (represented by Bio14, Bio17, and Bio18 variables) showed a high positive correlation with seed dormancy and germination niche breadth indexes (PC1 scores, GA, and GE, see Figure 3 and Table A6) when all genotypes of the specialist sexual Dilatata species were considered. This relationship suggests that precipitation, when seedling emergence and growth occurs (late spring and summer), may be a strong environmental filter driving local adaptation of seed dormancy and germination traits for these species. Due to the seedling morphology of panicoid grasses, adequate soil water availability near the soil surface is particularly important during seedling emergence [88,89]. A decreasing precipitation gradient often implies increasing seed dormancy across populations or species [46,90], however, there are some situations where the inverse seed dormancy–precipitation association was reported [91,92]. 

The distribution models for the two northernmost specialist species (*P. vacarianum* and *P. plurinerve*) yielded very close areas, even showing an overlapping area where both species got high favourability. Still, the central zone of each distribution area was different (Figure 1). *Paspalum vacarianum* occurs in the Brazilian Planalto (1000 m.a.s.l.), a region with a similar precipitation regime but with colder winters than in the distribution area of *P. plurinerve*. Although the seed dormancy and germination niche breadth indexes analysed were similar between these species, the germination in mildly cold temperatures (15/25 °C) was strongly inhibited for *P. vacarianum*, though not for *P. plurinerve* (Table 2). Species from higher altitudes and lower annual minimum temperatures tend to require higher minimum temperatures to germinate [48]. The inhibition of the germination of *P. vacarianum* in colder temperatures may suggest a local adaptation of seed germination to the climate of the Brazilian Planalto that differentiates it from *P. plurinerve*. 

## 4. Materials and Methods

### 4.1. Distribution Data 

This study includes the five sexual species of the Dilatata group: *Paspalum dasypleurum*, *P. flavescens*, *P. plurinerve*, *P. vacarianum*, and *P. urvillei*. Their occurrences were obtained from the literature and the examination of herbarium specimens deposited in BAA, BLA, CEN, CORD, CONC, CNPO, CTES, ICN, LIL, LP, MNES, MVFA, MVM, and SI (acronyms for Thiers [93]). The coordinates were taken from the labels; when there were none, they were georeferenced to the location specified in the label using Google Earth. In addition, for *P. dasypleurum* and *P. urvillei*, the Global Biodiversity Information Facility (GBIF, accessed in July 2020) was consulted, considering only the native area of each species [94,95,96,97]. Additionally, for these two species, all the herbarium specimens of BAA and CORD were seen and analysed and herbarium specimens from the other institutions were visualized in order to confirm species and location. For *P. flavescens*, *P. plurinerve*, and *P. vacarianum*, all specimens available in all herbaria were used [12]. Most of them were georeferenced using their field labels.

### 4.2. Species Distribution, Environmental Variables, and Favourability Function 

The analyses were carried out considering South America as the study area. The area was divided into 181,221 hexagons (6 km of apothem) (see Figure A1). We chose hexagons since they look more appropriate in connectivity studies, giving a better correspondence between the measured and Euclidian distances than rectangular grids [98]. From the distribution data, we obtained the presence and absence of each hexagon. Hexagons with at least one record were marked as a presence for the species, and the others with no record were marked as absences. We considered a total of 50 explanatory variables related to different environmental factors (spatial, topography, climatic, hydrology, global land cover, and human activities, see the list and sources in Table A1). The average value of each explanatory variable was obtained for each hexagon of the study grid. All these tasks were done using tools from QGIS v3.14 [99]. The spatial factor (Ysp) was built using a polynomial trend-surface analysis that includes quadratic, cubic, and interaction effects of latitude (La) and longitude (Lo) (Lo, Lo2, Lo3, La, La2, La3, LaLo, La2Lo, and LaLo2). This spatial descriptor detects geographic trends that are not evident with other environmental variables [21,100,101,102,103,104].

Based on the presence-absence data and environmental dataset, we optimized SDMs for each species according to the favourability function as the modelling algorithm [31]. Favourability values (*F*) can be obtained as follows;
(1)$$F=\frac{\left[\frac{P}{\left(1-P\right)}\right]}{\left[\frac{n_{1}}{n_{0}}+\left(\frac{P}{\left(1-P\right)}\right)\right]}\;$$
where *F* is the environmental favourability, *P* is the probability of occurrence obtained from the multivariate logistic regression, *n*_1_ and *n*_0_ are the numbers of presences, and absences, respectively. 

To obtain *F*, we used the ‘fuzzySim’ package [105] implemented in R [106]. The *multGLM* function was used, which allows the analytical procedures to be carried out sequentially in several steps. To minimize the effect of multicollinearity among the variables, two filters were applied to perform a preliminary variable selection from the initial set of 50 explanatory variables to use uncorrelated variables without problems of false discoveries. First, we controlled for a type I error using the false discovery rate (FDR) according to Benjamini and Hochberg [107] with the *FDR* function. Second, we calculated Pearson correlations among the variables using a threshold value of 0.8 with the *corSelect* function for the variables that passed the FDR filter. Using the variables that overcame both filters, models were constructed with the *step* function which performs a backward and forward step-by-step variable selection according to the Akaike information criterion (AIC) [108]. Finally, we used the *multTrim* function to remove nonsignificant variables. A Wald test was carried out using the ‘survey’ package [109] to determine the relative importance of each variable in all the models. In addition, we checked that all the explanatory variables selected for each model had a variance inflation factor lower than ten (VIF < 10), which is the threshold to indicate the absence of multicollinearity following Montgomery and Peck [110]. Finally, we used the ‘modEvA’ package [111] to evaluate the performance of the final models. To assess the prediction accuracy of the models, we calculated the following classification parameters: sensitivity, specificity, correct classification rate (CCR), and the true skill statistic (TSS) [112,113]. On the other hand, we estimated the AUC as the discrimination parameter [114]. 

### 4.3. Seed Production and Harvest

Based on the selfing breeding system of these species, the seed of each plant was considered a single inbred line. Eleven lines (genotypes) from the five sexual species of the Dilatata group were used to produce seeds (one genotype of *P. dasypleurum*, three of *P. flavescens*, and *P. urvillei*, and two of *P. plurinerve* and *P. vacarianum*, see Figure 1 and Table A3). Eight plants of each genotype were installed 1 m apart from one another in a common garden experiment in Montevideo, Uruguay (34°511 S, 56°120 W). Seeds (spikelets with caryopses) of the eight plants of each genotype were bulk harvested by hand threshing for two weeks at each harvest time: in December 2017 and 2018 (summer), and March 2018 and 2019 (fall). The harvested seeds were kept in paper envelopes and stored in a dry place at room temperature for one week. Then, they were put in airtight bags with silica gel at 6 °C to retain primary seed dormancy until used in germination experiments [115,116]. We evaluated the germination of seeds with primary dormancy to reduce environmental postharvest effects on germination phenotypes and to compare genotypes and harvest times in a more reliable way.

### 4.4. Germination Experiments

For each harvest time, two completely randomized germination experiments were carried out. In each experiment, three replicates of thirty to fifty seeds for each genotype and treatment were placed in Petri dishes on filter paper moistened with 5 mL of distilled water or a nitrate solution (0.2% *w/v* KNO_3_). The Petri dishes were wrapped in film to avoid loss of humidity. A seed was considered germinated when a 1 mm or further radicle growth was visible. Nongerminated seeds were tested with the tetrazolium test following Maeda et al. [117] to determine the number of viable seeds in each Petri dish after the germination assay had elapsed.

Experiment one was set to assess germination after stratification treatments in order to quantify differences among genotypes in thermal requirements to break seed dormancy. Three treatments were conducted. (1) Seeds with cold stratification (7 d in 9 °C average), (2) with warm stratification (7 d in 20 °C) and nonstratified seeds. The Petri dishes with seeds moistened with distilled water were placed in a refrigerator for cold stratification, and in a controlled temperature chamber for warm stratification. The dishes were wrapped with dark nylon to avoid light during the stratification period. After stratification, the dishes were subjected to two consecutive incubation periods: (1) in germination chambers at constant 30 °C for four days, and (2) in germination chambers with alternating temperatures (20/30 °C, 12 h dark/ 12 h light) for seven additional days. The dishes with nonstratified seeds were subjected to the same germination conditions. Germination counts were done on the 2nd, 3rd and 4th days during the incubation period in steady temperature, and on the 2nd, 4th and 7th days during the period in alternating temperatures.

Experiment two was set to assess germination in a range of alternating temperatures with and without nitrate to determine the thermal amplitude of germination in the presence or not of a dormancy-breaking substance. Four alternating temperature regimes (10/20, 15/25, 20/30 and 25/35 °C, 12 h dark/12 h light) and two germination solutions (distilled water and 0.2% nitrate solution) were factorialised. Four germination chambers were used simultaneously, each with one alternating temperature regime. The time to finalize the germination assay was different for each alternating temperature regime to allow similar thermal time accumulation (28 d for 10/20 °C, 21 d for 15/25 °C, 17 d for 20/30 °C and 14 d for 25/35 °C). The germination counts were done at intervals of two to three days until germination assay time elapsed.

### 4.5. Germination Variables 

For each replicate, the final germination proportion (FGP), germination timing variables, and the uncertainty of germination (UG) of both experiments were estimated. The FGP was calculated as the number of germinated seeds divided by the number of viable seeds (germinated + positive in the tetrazolium test). The mean germination time (MGT) was estimated for experiment two as;
(2)$$\mathrm{MGT}=\frac{\sum_{i}^{k}g_{i}{\times t}_{i}}{\sum_{i}^{k}g_{i}}$$
where *g_i_* is the number of germinated seeds in count *i* and *t_i_* is the time in days to count *i* since the beginning of the experiment. For experiment one, we defined a germination section as the time interval between counts, and we estimated the mean germination section (MGS) using a modified Timson Index (Tmod) [118];
(3)$$\mathrm{MGS}=k-\mathrm{Tmod}=k-\left[\frac{\sum_{i}^{k}g_{i}\times \left(k-h\right)}{\sum_{i}^{k}g_{i}}\right]$$
where *g_i_* is the number of germinated seeds in count *i*, *k* is the total number of germination counts (*k* = 6), and *h* = *i*−1. The MGS brings an idea of mean germination time when the incubation of germination includes condition changes. The UG measured the spreading of germination which was used to infer the synchronicity of germination. It was estimated following Marques et al. [54] as;
(4)$$\mathrm{UG}=-\left[\overset{k}{\underset{i}{\sum }}f_{i}\times \log_{2}\left(f_{i}\right)\right],f_{i}=\frac{g_{i}}{\sum_{i}^{k}g_{i}}$$
where *g_i_* is the number of germinated seeds in count *i*, *k* is the total number of germination counts.

Using the mean FGP obtained at each treatment level of both experiments, we estimated variables that can account for the overall germination performance of each genotype and harvest time. The germination ability (GA) was calculated as the summation of all FGP obtained through the treatment levels of both experiments (maximum GA = 11). The germination evenness (GE) across treatments was calculated using a modified Levin’s index, following Finch et al. [53], as;
(5)$$\mathrm{GE}=\frac{1}{R\times \sum_{j}^{R}{\left(p_{j}\right)}^{2}}$$
where *R* is the total number of treatment levels (*j*) of both experiments (*R* = 3 and *R* = 8 for experiments one and two, respectively. Total *R* = 11) and *p_j_* is the proportion of germinated seeds of level *j* in a base of all germinated seeds (*p_j_* = FGP*_j_*/GA). Both GA and GE can be used as germination niche breadth and seed dormancy estimators.

### 4.6. Germination Data Analysis

Variance analysis for the germination variables was conducted using mixed-effect models to find significant differences among species and treatments. The analysis was performed with the *lmer* function from the ‘lme4’ package [119]. Harvest time and genotype-by-species interaction were assumed as random effects. Species and germination treatments were the fixed effects. The FGP and GE data were logit transformed before the analysis to meet normal distribution. For GA and GE, the treatment effect was omitted from the models. Adjusted means for all fixed effects were calculated, and pairwise comparisons among factor levels were made with Tukey’s test (α = 0.05). 

A principal component analysis (PCA) was carried out to study germination data dispersion using the FGP, MGT (or MGS), and UG obtained in each treatment level of both experiments as quantitative variables for each genotype and harvest time. Species and genotypes were considered categorical supplementary variables. The germination variables have different scales, so the data were normalized. The PCA was performed with the *PCA* function of the ‘FactoMineR’ package [120]. 

To assess the relative importance of within and among species variance, the variance partitioning was estimated for the germination variables by mixed effect models with harvest time, species, and genotype-by-species as random effects. The significance of random effects was assessed by the likelihood ratio test.

### 4.7. Regressions between Germination and Environmental Variables

Partial least square regressions (PLSR) were conducted to find relationships between germination and environment variables. We used the mean values of germination variables of each genotype (responses) and the mean values of each climatic and topographic variable inside the hexagon where each genotype is located (predictors). The PLSR was carried out with the *plsr* function of the ‘pls’ package [121]. The leave-one-out cross-validation process was used to retain an adequate number of components for each PLSR. For each germination variable, two PLSR were conducted. All variables were used as predictors in the first one, and the variables with importance in projection higher than one (VIP > 1) in the first PLSR were used in the second. The VIP of each variable was estimated with *VIP* function of ‘plsVarSel’ package [122]. Then, linear regressions were tested for each germination variable using the environmental variables with VIP > 1 after the last PLSR as the predictor.

## 5. Conclusions

We analysed and compared distribution models and regeneration traits in one phylogenetically well-understood group of closely-related species from a region where biogeographical studies are scarce. Each methodological approach brought different and complementary insights. While the SDMs yielded the favourable area of each species, highlighting the main drivers of the distributions, the differences among species in germination traits provided a likely physiological explanation for the geographical pattern of each species and the whole group. The divergence in seed dormancy observed among *P. flavescens*, *P. plurinerve*, and *P. vacarianum* may be a key factor in explaining why these more closely-related species retain restricted and allopatric distributions. This divergence gives each of these species a competitive advantage in the regeneration niche in their location over the other specialist sexual Dilatata species. 

## Figures and Tables

**Figure 1 plants-12-01342-f001:**
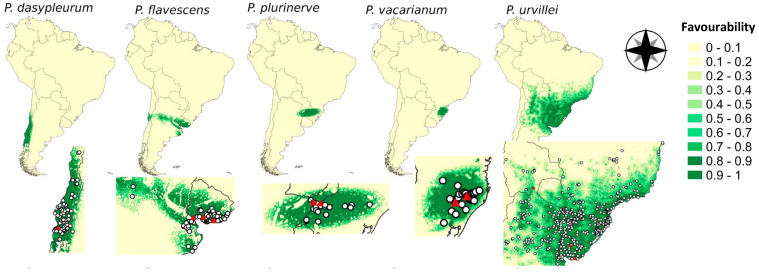
Favourability models for each of the five sexual species of the Dilatata group based on predictor variables from different factors: spatial configuration, topography, climatic, hydrology, global land cover, and human activities (see Table A1). The zoom of each favourability model (bottom maps) shows the species occurrences (white circles) and the occurrence of genotypes used for germination assays (red triangles).

**Figure 2 plants-12-01342-f002:**
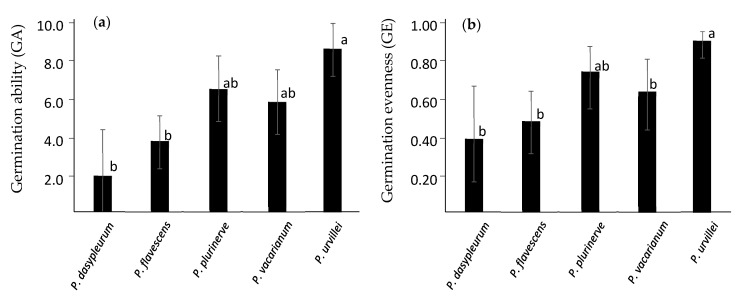
(**a**) Mean germination ability (GA) and (**b**) mean germination evenness (GE) for each species. These variables were generated to estimate both seed dormancy and germination niche breadth. GE data were logit transformed before analysis. Vertical lines indicate a 95% confidence interval. Different letters mean significant differences among species within each germination variable.

**Figure 3 plants-12-01342-f003:**
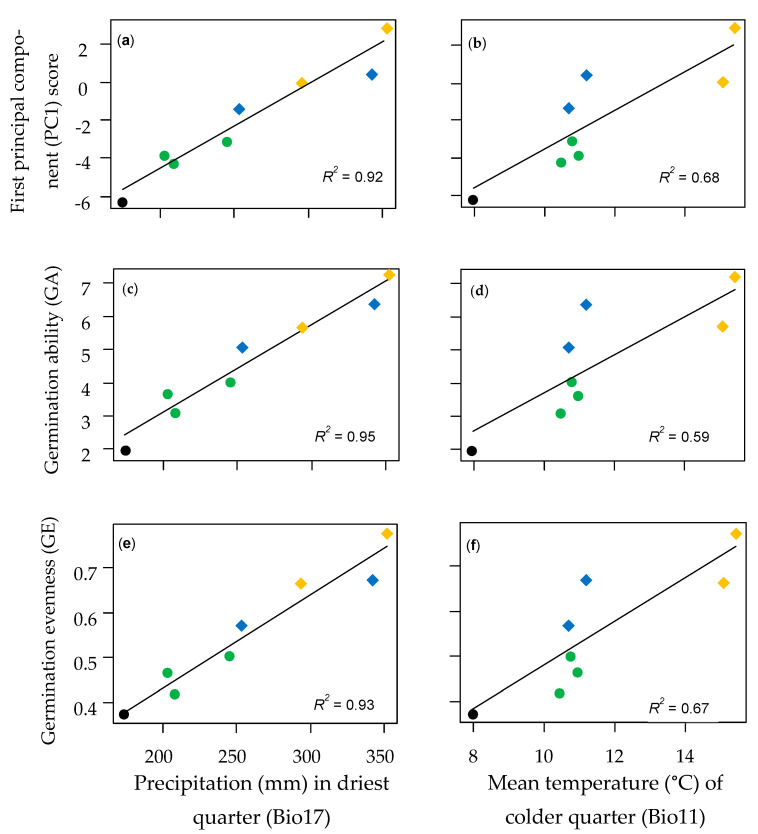
Linear regressions with precipitation in the driest quarter (**a**,**c**,**e**) and with the mean temperature of the colder quarter (**b**,**d**,**f**) for three germination variables: First principal component (PC1) score from PCA (**a**,**b**), germination ability (GA) (**c**,**d**), and germination evenness (GE) (**e**,**f**). Colours identify the species: black = *P. dasypleurum*, green = *P. flavescens*, blue = *P. vacarianum*, and orange = *P. plurinerve*. Genotypes of *P. urvillei* were not included. Southernmost distributed species are represented with circles and northernmost species with diamonds. Mean values for each genotype of the four species were used. The determination coefficient (*R^2^*) for each regression is reported.

**Table 1 plants-12-01342-t001:** Explanatory variables, and factors they belong to, for each of the five sexual species of the Dilatata group in the SDMs obtained. Wald parameter values (according to Wald’s test) for predictor variables included in each favourability model indicate the relative importance of each variable. Bold numbers highlight the four most significant predictor variables of each SDM. Signs in brackets show the positive or negative relationship between favourability and the variable. Abbreviations and sources of each predictor variable are noted in Table A1.

Factors	Variables	*P. dasypleurum*	*P. flavescens*	*P. plurinerve*	*P. vacarianum*	*P. urvillei*
Spatial	Ysp	12.53 (+)	107.32 (+)	19.65 (+)	11.40 (+)	81.09 (+)
Topography	Slope	18.80 (+)				6.14 (+)
Climatic	Bio3 (isothermality)					18.20 (−)
	Bio6 (Tmin coldest month)				18.98 (−)	
	Bio9 (Tmean driest quarter)	9.17 (+)				8.75 (−)
	Bio14 (Precip. driest month)					10.80 (+)
	Bio19 (Precip. coldest quarter)					7.97 (−)
	ClimMoist	17.88 (+)				
Hydrology	DistBigRiv		10.15 (+)			13.77 (−)
Global land cover	Urban	38.59 (+)			10.19 (+)	82.30 (+)
	Crops					36.83 (−)
	Bare					8.39 (−)
	Shrub					8.61 (−)
	WaterSeas		7.36 (−)			
Human activities	DistRoad	10.75 (−)	7.39 (−)	5.76 (−)	8.39 (−)	
	PopDen		7.90 (+)			
	DistUrban					59.26 (−)
	SumRoad					76.07 (+)

**Table 2 plants-12-01342-t002:** Adjusted means for final germination proportion (FGP) for each level of treatment and species within each germination experiment. (**a**) Experiment one (stratification treatments), (**b**) Experiment two (alternating temperatures and nitrate addition).

(a) Treatment	*P. dasypleurum*			*P. flavescens*			*P. plurinerve*			*P. vacarianum*			*P. urvillei*		
without stratification	0.04	a	B	0.04	c	B	0.54	c	AB	0.38	b	AB	0.86	b	A
cold stratification	0.08	a	C	0.49	a	BC	0.96	a	AB	0.86	a	AB	0.97	a	A
warm stratification	0.04	a	B	0.13	b	B	0.83	b	AB	0.50	b	AB	0.96	a	A
**(b) Treatment**	** *P. dasypleurum* **			** *P. flavescens* **			** *P. plurinerve* **			** *P. vacarianum* **			** *P. urvillei* **		
10/20 °C water	0	a	A	0	b	A	0	b	A	0	b	A	0.04	b	A
15/25 °C water	0	a	B	0.03	b	B	0.21	a	AB	0.05	b	B	0.75	a	A
20/30 °C water	0.11	a	B	0.31	a	B	0.48	a	AB	0.89	a	A	0.83	a	A
25/35 °C water	0.19	a	B	0.37	a	B	0.39	a	B	0.91	a	A	0.78	a	AB
10/20 °C nitrate	0	b	B	0.01	b	B	0.10	b	B	0.02	c	B	0.81	b	A
15/25 °C nitrate	0.12	ab	B	0.10	b	B	0.98	a	A	0.41	b	B	0.98	a	A
20/30 °C nitrate	0.39	ab	B	0.96	a	A	0.99	a	A	0.98	a	A	0.99	a	A
25/35 °C nitrate	0.54	a	B	0.98	a	A	0.99	a	A	0.98	a	A	0.99	a	A

(**a**) Lowercase letters indicate differences among stratification treatments within each species. Uppercase letters indicate differences among species within each stratification treatment (*p* < 0.05). (**b**) Lowercase letters indicate differences among temperature regimes within each species and nitrate addition combination. Uppercase letters indicate differences among species within each temperature regime and nitrate addition combination. Bold numbers indicate differences between distilled water and nitrate addition within each temperature regime and species combinations (*p* < 0.05).

**Table 3 plants-12-01342-t003:** Optimization results of partial least square regressions (PLSR) were performed for each response (germination) variable. (**a**) Optimization including *P. urvillei* genotypes. (**b**) Optimization without *P. urvillei* genotypes. Root mean squared error on prediction (RMSEP) for the selected number of components and percentage of predictor (environmental variables) and response (germination variables) explained by PLSRs are shown. The predictor variables with importance in projection higher than one (VIP > 1) were listed for each PLSR optimization. Abbreviations and sources of each predictor variable are noted in Table A1.

(a) Including *P. urvillei*					
**Response** **Variable**	**N° of** **Components**	**RMSEP**	**% Predictor Variance Explained**	**% Response Variance Explained**	**Variables with Importance in** **Projection (VIP > 1)**
PC1 scores from PCA	2	2.505	79.4	61.7	Bio1, Bio2, Bio5, Bio11, ETP_An, ClimMoistu, GDD0, OriNS
GA (germination ability)	2	1.405	79.8	61.7	Bio1, Bio2, Bio11, ETP_An, GDD0, OriNS
GE (germination evenness)	2	0.123	81.9	55.7	Bio1, Bio2, Bio11, ETP_An, GDD0
**(b) Without *P. urvillei***					
**Response** **Variable**	**N° of** **Components**	**RMSEP**	**% Predictor Variance Explained**	**% Response Variance Explained**	**Variables with Importance in** **Projection (VIP > 1)**
PC1 scores from PCA	2	0.560	94.4	94.9	Bio2, Bio11, Bio14, Bio17, Bio18, ETP_An
GA (germination ability)	2	0.349	87.5	95.6	Bio11, Bio14, Bio17, Bio18, ETP_An
GE (germination evenness)	2	0.028	94.5	95.4	Bio2, Bio11, Bio14, Bio17, Bio18, ETP_An

## Data Availability

The data presented in this study are openly available in Zenodo. DOI: 10.5281/zenodo.7472847.

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
