# Peer review of "Understanding the Geographic Patterns of Closely-Related Species of Paspalum (Poaceae) Using Distribution Modelling and Seed Germination Traits"

_plants, 2023, doi:10.3390/plants12061342_

Round 1
Reviewer 1 Report
This work constitutes an interesting contribution to the interpretation of the distribution patterns of five taxa of the genus Paspalum (Dilatata group), combining the elaboration of species distribution models (SDM) with the germination characteristics of the species studied.
Overall, I think the manuscript is clear, well structured and carefully written (I found almost no typing errors). The interpretation of the results seemed good, although sometimes difficult to follow given the large number of analyses done (would it not be possible to limit the analyses presented to those that are really important?).
I think the title does not reflect the work done, as no interpretation of the biogeographical pattern is given. Such an analysis would require a broader approach. I, therefore, suggest that it be adjusted to reflect more clearly the work done.
Ln 137 – ‘The spatial variable (Ysp) was relevant across the models of the five species, and it was the most important predictor for the distribution of P. flavescens and P. plurinerve (Table 1).’ – The use of the spatial factor (Ysp) as an explanatory variable raises some doubts to me, since species do not respond to geographical coordinates, but to the climatic factors associated with them. Similarly, the role of the variable "urban cover" (e.g. also "Urban cover as the most relevant predictor for P. dasypleurum and P. urvillei") seems strange.
It is essential to verify the possible existence of multicollinearity between the selected explanatory variables. Was it carried out? I did not find any reference in the manuscript.
Ln 157 – In Table 1, bold numbers (referred to in the legend) are missing.
Ln 168, 271, 445 – ‘dormancy-braking’ – replace by ‘dormancy-breaking’
Ln 193 – ‘varia-bility’ – replace by ‘variability’
Ln 197 – ‘The PC1 scores for each genotype and harvest time summarized the values of the estimated germination variables for all treatments in both experiments; therefore, it was considered another variable measuring seed dormancy.’ – I don't understand what this paragraph means. What was considered another variable?
Ln 206 – Despite small differences, the genotypes of P. urvillei showed high germination under most conditions assayed even in low temperatures (10/20 °C) (Table 2, Figure A2 and A3), regardless of the large distances among collection sites. Thus, the germination response of P. urvillei did not appear to be strongly associated with an environmental gradient. – This paragraph, in particular the second sentence, seems confusing and even misleading to me. Please check what is meant to be said.
Ln 254 – ‘The distribution models of each species yielded a high favourability area which is highly coincident with the observed distribution range of each species.’ – Isn't this to be expected?!
Ln 277 – ‘Nitrate in the soil is considered an environmental cue for the absence of adult plants.’ – What is meant by this sentence?
Table A1. – The solid line separating the types of variables in the two columns is confusing (e.g. the line before 'Hydrology' or before 'Human activities'). Lines should be broken to make it clear that the information is in two columns.
Ln 410 – It is not clear what is the total number of plants analysed. In the text, it is stated: "Eight plants of each genotype". Does this mean that a total of 88 plants (11x8) were used? With the species to be sampled differently (e.g., 8 for P. dasypleurum and 24 for P. flavescens)?
Author Response
Please, see the attachment

Reviewer 2 Report
General comments:In this manuscript the authors propose a very accurate, clear and precise study on species distribution models (SDMs) integrated with seed germination in order to determine whether germination divergences explain biogeographic patterns.
I found this research work very interesting and from my point of view, the idea fits perfectly within the scope of the journal.
In addition, the aims of the paper are consistent with the results and the discussion.
The volume of information that the authors have managed is remarkable, and the references are recent and adequate.
Few specific comments:
Line 43: Change in the absence with and the absence
Lines 52-54: This sentence doesn’t sound right.Please, change with: Despite their close phylogenetic relationship, the five sexual species in this group can be considered very well-delimited and recently diverged independent evolutionary units [14, 15].
Author Response
Please, see the attachment

Reviewer 3 Report
Good job,
your interesting manuscript entitled: "Understanding biogeographical patterns of closely related species of Paspalum (Poaceae) using distribution modeling and seed germination traits" analyzed and compared distribution models and regeneration traits in a phylogenetically well-understood group of closely related species from an area where biogeographical studies are limited. The distribution of a plant species is determined by historical and environmental factors and its interaction with other species. Species distribution models (SDMs) have been used extensively as an approach to understand the potential distribution of a species and to test biogeographical, ecological and evolutionary hypotheses. As noted, the divergence in seed dormancy observed among P. flavescens, P. plurinerve and P. vacarianum may be a key factor in explaining why these more closely-related species retain restricted and allopatric distributions. This divergence gives each of these species a competitive advantage in the regeneration niche in their location over the other specialist sexual Dilatata species. Adequately, you assessed differences in seed dormancy between species and performed regressions between vegetation and environmental variables for each population. Finally, the ecological and biogeographical insights brought by each approach were adequately discussed.
· Correction 1: In Table 1. Line 160, Bold numbers highlight the four most … there are no bold numbers in the table
Author Response
Please, see the attachment
